# Exploring the Camera Bias of Person Re-identification

**Myungseo Song & Jin-Woo Park**
mAy-I Inc.
Seoul, Korea
{myungseo.song,jin}@may-i.io

**Jong-Seok Lee**
Yonsei University
Seoul, Korea
jong-seok.lee@yonsei.ac.kr

## Abstract

We empirically investigate the camera bias of person re-identification (ReID) models. Previously, camera-aware methods have been proposed to address this issue, but they are largely confined to training domains of the models. We measure the camera bias of ReID models on unseen domains and reveal that camera bias becomes more pronounced under data distribution shifts. As a debiasing method for unseen domain data, we revisit feature normalization on embedding vectors. While the normalization has been used as a straightforward solution, its underlying causes and broader applicability remain unexplored. We analyze why this simple method is effective at reducing bias and show that it can be applied to detailed bias factors such as low-level image properties and body angle. Furthermore, we validate its generalizability across various models and benchmarks, highlighting its potential as a simple yet effective test-time postprocessing method for ReID. In addition, we explore the inherent risk of camera bias in unsupervised learning of ReID models. The unsupervised models remain highly biased towards camera labels even for seen domain data, indicating substantial room for improvement. Based on observations of the negative impact of camera-biased pseudo labels on training, we suggest simple training strategies to mitigate the bias. By applying these strategies to existing unsupervised learning algorithms, we show that significant performance improvements can be achieved with minor modifications.

## 1 Introduction

Person re-identification (ReID) is a process of retrieving images of a query identity from gallery images. With recent advances in deep learning, a wide range of challenging ReID scenarios have been covered, including object occlusion (Miao et al., 2019; Somers et al., 2023), change of appearance (Jin et al., 2022), and infrared images (Wu et al., 2017; Wu & Ye, 2023). In general, the inter-camera sample matching is not trivial since the shared information among images from the same camera can mislead a model easily. This phenomenon is known as the problem of camera bias, where samples from the same camera tend to gather closer in the feature space. This increases the false matching between the query-gallery samples since the samples of different identities from the same camera can be considered too similar. To address the issue, camera-aware ReID methods (Luo et al., 2020; Wang et al., 2021; Chen et al., 2021; Cho et al., 2022; Lee et al., 2023) have been proposed, aiming to learn camera-invariant representations by leveraging camera labels of samples during training.

However, the previous works on camera bias of ReID models have mainly focused on seen domains of the models, while the camera bias of ReID models on unseen domains has been overlooked. We observe that existing ReID models exhibit a large camera bias for unseen domain data. For example, Figure 1 describes the feature distance distributions between samples of a camera-aware model (Cho et al., 2022) trained on the Market-1501 (Zheng et al., 2015) dataset, using samples from the MSMT17 (Wei et al., 2018) dataset. Compared to the distance distributions of the seen domain samples, the distance distributions of the unseen domain samples are more separable.

In this paper, we first investigate the camera bias of existing ReID models on seen and unseen domain data. We observe that, regardless of the model types, there is a large camera bias in distribution shifts, and unsupervised models are vulnerable to camera bias even on seen domains. As a straightforward

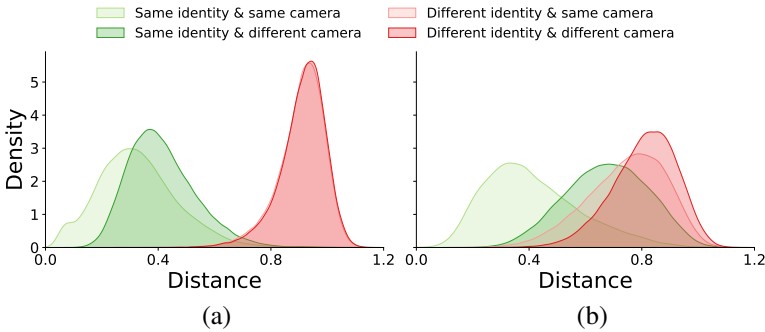

Figure 1: Cosine distance distributions of a camera-aware ReID model on (a) the training domain (Market-1501) and (b) the unseen domain (MSMT17). The distances between samples within the same cameras are more skewed to the left when the data distribution is shifted.

debiasing technique for unseen domains, we revisit the normalization method on the embedding features of ReID models. Through comprehensive empirical analysis, we reveal why the feature normalization effectively reduces biases towards camera labels and fine-grained factors such as low-level image properties and body angles, as well as demonstrating its general applicability for various ReID models. Additionally, we explore the inherent risk of camera bias in unsupervised learning (USL) of ReID models, observing the negative impact of camera-biased pseudo labels on training. Based on our analysis, we suggest simple training strategies applicable to existing USL algorithms, which significantly improve the performance.

The main contributions of this work are summarized as follows:

- We investigate the camera bias of ReID models on unseen domain data, which has not been thoroughly studied. We provide comprehensive analysis encompassing various learning methods and model architectures.

- We revisit the debiasing effects of normalization on embedding vectors of ReID models. The empirical analysis explains why it is effective for bias mitigation and shows its applicability to detailed bias factors and multiple models.

- We explore the risk of camera bias inherent in unsupervised learning of ReID models. From this, we show that the performance of existing unsupervised algorithms can be effectively enhanced by simple modifications to reduce the risk.

## 2 RELATED WORK

In traditional person ReID methods, the convolutional neural networks (CNN) architectures have been popularly adopted with cross-entropy and triplet loss (Zheng et al., 2017; Hermans et al., 2017; Luo et al., 2019; Ye et al., 2021). When identity labels of training data are unavailable, the pseudo labels are used instead based on clustering on the extracted features (Fan et al., 2018; Lin et al., 2019; Yu et al., 2019; Zhang et al., 2019; Dai et al., 2022). Recently, the transformer backbones (He et al., 2021; Luo et al., 2021b; Chen et al., 2023) and self-supervised pretraining (Fu et al., 2021; 2022; Luo et al., 2021b; Chen et al., 2023) significantly improve the ReID performance. To enhance the generalization ability of the models, a variety of domain generalizable techniques have been also proposed (Dai et al., 2021; Song et al., 2019; Liao & Shao, 2021; Ni et al., 2023; Dou et al., 2023).

However, it has been found that the ReID models are biased towards the camera views of given data. The camera-aware methods have been proposed to alleviate this problem, where camera labels of the samples are utilized in model training as auxiliary information (Luo et al., 2020; Zhuang et al., 2020; Zhang et al., 2021; Wang et al., 2021; Chen et al., 2021; Cho et al., 2022; Lee et al., 2023). For example, an inter-camera contrastive loss is proposed to minimize the variations of the features from different cameras within the same class (Wang et al., 2021; Cho et al., 2022). Zhuang et al. (2020) replace batch normalization layers of a model with camera-based batch normalization layers conditional to the camera labels of inputs to reduce the distribution gap. Some other studies (Gu et al., 2020; Luo et al., 2021a) post-process a feature by subtracting the mean feature within its camera

Table 1: Camera bias and accuracy of various state-of-the-art ReID models based on clustering results. "SL" and "CA" denote the supervised learning and camera-aware method, respectively. "Bias" and "Accuracy" denote the Normalized Mutual Information (NMI) scores between cluster labels and camera labels, and between cluster labels and identity labels, respectively, in $\times 100$ scale. ISR is trained on external videos and the other models are trained on MSMT17-Train.

| Method | SL | CA | Backbone | MSMT17-Train | | MSMT17-Test | | Market-1501 | | CUHK03-NP | | PersonX | |
|---|---|---|---|---|---|---|---|---|---|---|---|---|---|
| | | | | Bias | Accuracy | Bias | Accuracy | Bias | Accuracy | Bias | Accuracy | Bias | Accuracy |
| CC (Dai et al., 2022) | ✗ | ✗ | R50 | 34.7 | 89.3 | 32.5 | 88.0 | 17.1 | 81.0 | 17.6 | 74.6 | 20.6 | 78.9 |
| PPLR (Cho et al., 2022) | ✗ | ✗ | R50 | 31.8 | 90.3 | 30.2 | 89.0 | 15.6 | 81.7 | 15.9 | 77.4 | 15.3 | 82.0 |
| TransReID-SSL (Luo et al., 2021b) | ✗ | ✗ | ViT | 29.3 | 93.1 | 27.1 | 92.8 | 9.7 | 92.2 | 7.0 | 84.2 | 12.5 | 88.8 |
| ISR (Dou et al., 2023) | ✗ | ✗ | ViT | 31.8 | 90.5 | 30.3 | 89.4 | 9.7 | 95.8 | 5.4 | 87.7 | 6.1 | 94.9 |
| PPLR-CAM (Cho et al., 2022) | ✗ | ✓ | R50 | 29.3 | 92.8 | 26.7 | 92.4 | 14.3 | 84.1 | 13.7 | 78.4 | 14.6 | 81.8 |
| TransReID (He et al., 2021) | ✓ | ✓ | ViT | 24.4 | 98.3 | 23.6 | 94.5 | 13.6 | 89.8 | 3.9 | 84.7 | 6.6 | 92.7 |
| SOLIDER (Chen et al., 2023) | ✓ | ✗ | ViT | 23.2 | 98.7 | 21.3 | 96.9 | 7.3 | 96.5 | 1.6 | 90.8 | 2.8 | 93.8 |
| Ground Truth | - | - | - | 21.1 | - | 19.2 | - | 6.4 | - | 0.1 | - | 0.0 | - |

view, but this is performed without justification and is limited to an unsupervised domain adaptation task. These previous studies have primarily focused on the bias of the models on the training domain data, while the bias on unseen domain data has been neglected. Meanwhile, we call the methods which do not take the camera views into account camera-agnostic methods.

# 3 QUANTITATIVE ANALYSIS ON CAMERA BIAS

In this section, we quantitatively investigate the camera bias in existing ReID models. The camera bias is the phenomenon where the feature distribution is biased towards the camera labels of the samples, which degrades ReID performance. Many camera-aware methods have been proposed to address this problem. However, the scope of the discussion has been primarily limited to training domain and the camera bias on unseen domains has not been thoroughly explored. We focus on the camera bias of ReID models on unseen domains, examining various types of models including camera-aware/agnostic, supervised/unsupervised, and domain generalizable approaches, with the widely used backbones such as ResNet (He et al., 2016) and ViT (Dosovitskiy et al., 2021).

To measure the bias, we utilize Normalized Mutual Information (NMI) which quantifies the shared information between two clustering results. We extract the features of samples and perform clustering to them using InfoMAP (Rosvall & Bergstrom, 2008). Then, the camera bias is computed by NMI between cluster labels and camera labels of the samples. The accuracy of the clusters are measured by NMI between the cluster labels and the identity labels. The results on MSMT17, Market-1501, CUHK03-NP (Zhong et al., 2017a), and PersonX (Sun & Zheng, 2019) are shown in Table 1, where the bias of the ground truth (*i.e.*, NMI between the identity labels and the camera labels) indicates the inherent imbalance in a dataset. All models except ISR (Dou et al., 2023) are trained on MSMT17, hence the other datasets are unseen domains for them. For ISR, all datasets are unseen domains.

We make two notable observations from the results. First, the existing ReID models have a large camera bias on the unseen domains, regardless of their training setups or backbones. Second, the unsupervised models have a large camera bias on the seen domain, even on their training data. These imply that debiasing methods for unseen domains are needed in general, and there is room for performance improvement of unsupervised methods by reducing the camera bias during training. Relatively, the recent supervised models exhibit less debiased results on the training domain.

# 4 UNDERSTANDING CAMERA BIAS AND FEATURE NORMALIZATION

## 4.1 CAMERA-SPECIFIC FEATURE NORMALIZATION

In Section 3, we observed that the ReID models have a large camera bias on unseen domains. As a straightforward debiasing method, we introduce camera-specific feature normalization which postprocesses embedding vectors leveraging camera labels at test time. It is performed as follows.

Suppose that a test dataset $\mathcal{X} = \{(\boldsymbol{x}_1, \boldsymbol{y}_1), (\boldsymbol{x}_2, \boldsymbol{y}_2), \cdots, (\boldsymbol{x}_N, \boldsymbol{y}_N)\}$ with $N$ samples is given, where $\boldsymbol{x}_i$ and $\boldsymbol{y}_i$ denote the image and camera label of each sample, respectively. A pretrained encoder

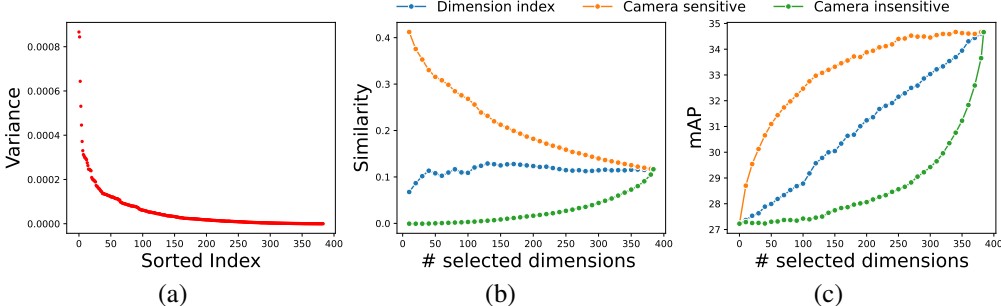

Figure 2: Analysis on the 384-dimensional embedding space of a ReID model. We measure the similarity of displacement vectors and mAP results increasing the number of feature dimensions following different orders. (a) Variance of each dimension of camera mean features. (b) Cosine similarity of displacement vectors between samples of the same identities from different cameras along selected dimensions. (c) Result of camera-specific feature centering for selected dimensions.

$f_\theta$ is used to extract embedding features $\mathcal{F} = \{\boldsymbol{f}_1, \boldsymbol{f}_2, \cdots, \boldsymbol{f}_N\}$, where $\boldsymbol{f}_i = f_\theta(\boldsymbol{x}_i)$. We split $\mathcal{F}$ into $M$ subsets $\mathcal{F}_1, \mathcal{F}_2, \cdots,$ and $\mathcal{F}_M$ depending on the camera labels, where the number of cameras is denoted by $M$. Then, the mean and standard deviation vectors for each camera, $\boldsymbol{m}_c$ and $\boldsymbol{\sigma}_c$, are computed as follows:

$$\boldsymbol{m}_c = \frac{1}{|\mathcal{F}_c|} \sum_{\boldsymbol{f}_i \in \mathcal{F}_c} \boldsymbol{f}_i \quad \text{and} \quad \boldsymbol{\sigma}_c = \sqrt{\frac{1}{|\mathcal{F}_c|} \sum_{\boldsymbol{f}_i \in \mathcal{F}_c} (\boldsymbol{f}_i - \boldsymbol{m}_c) \odot (\boldsymbol{f}_i - \boldsymbol{m}_c)}, \quad (1)$$

where $\odot$ denotes the element-wise multiplication. The camera-specific feature normalization on $\boldsymbol{f}_i$ with the camera label $\boldsymbol{y}_i$ is given by:

$$\hat{\boldsymbol{f}}_i = \frac{\boldsymbol{f}_i - \boldsymbol{m}_{\boldsymbol{y}_i}}{\boldsymbol{\sigma}_{\boldsymbol{y}_i}}. \quad (2)$$

This operation has been used as modified forms in camera mean subtraction (Gu et al., 2020; Luo et al., 2021a) and camera-specific batch normalization (Zhuang et al., 2020). In Zhuang et al. (2020), the normalization is followed by an affine transformation learned during training. However, how does the simple camera-specific feature normalization have a debiasing effect? We revisit the camera-specific feature normalization by empirically analyzing why it mitigates the camera bias and demonstrating its generalizability through comprehensive experiments.

## 4.2 ANALYSIS ON FEATURE SPACE

We dive deeply into the feature space of a ReID model (Luo et al., 2021b) trained on MSMT17 using CUHK03-NP samples, to understand why the normalization can play a role of debiasing.

**Sensitivity to camera variations differs across dimensions** We first find that the sensitivity of each dimension of the feature space to camera variations is quite different from each other. We compute mean features of each camera view and present the element-wise variances of the mean features in the descending order in Figure 2(a). It is shown that some dimensions have a relatively large variation, which might be largely related to the camera bias of the model.

**Movements of features due to camera variations** We indirectly investigate features, movements due to camera changes using the identity labels and camera labels of the samples. We obtain displacement vectors from feature pairs of two different cameras with the same identities (details in Appendix B.1) and compute their average cosine similarity in selected dimensions, with increasing the number of selected dimensions. Three selecting orders are used: (1) "Dimension index" follows the original index order of the dimensions, (2) "Camera sensitive" follows the descending order of the element-wise variances of the camera means, and (3) "Camera insensitive" follows the reverse order of (2). From Figure 2(b), we observe that the similarities of the displacement vectors in the camera-sensitive dimensions are relatively large. In other words, the features tend to move consistently in the

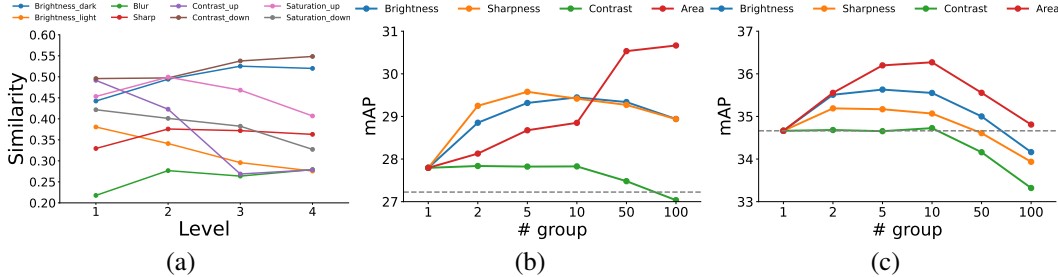

(a)  (b)  (c)

Figure 3: Analysis on low-level properties. (a) Cosine similarity of displacement vectors by image transformations. (b) Property group-specific feature normalization. The dashed line indicates the performance without normalization. (c) (Property group, camera)-specific feature normalization. The dashed line indicates the performance with camera-specific (and property-agnostic) normalization.

camera-sensitive dimensions depending on a camera variation, implying that the effect of a camera change appears as translation on these embedding dimensions.

**Sensitive dimensions dominate debiasing effects**   Then, can we debias the features by subtracting the camera mean features for those sensitive dimensions? To find out, we apply a camera-specific centering on selected dimensions in Figure 2(c). Note that there is a clear difference in the improvement rate of ReID performance depending on the selecting order. The performance gains are actually dominated in the camera-sensitive dimensions. For example, centering on top-50 dimensions (about 13%) of higher variances achieves approximately a half of the total gains, while centering on top-50 dimensions of lower variances shows almost no gain. For the low-variance dimensions, a half of the total gains requires centering of as many as 350 dimensions (about 91%). Similar results are obtained for other models in Appendix B.2.

### 4.3 ANALYSIS ON DETAILED BIAS FACTORS

We explore the feature normalization for detailed bias factors of ReID models, including image properties and body angle of images. The model (Luo et al., 2021b) trained on MSMT17 is used.

**Movements of features due to image transformations**   Given the fine-grained nature of person ReID, the camera bias of a model might be closely related to the difference in low-level image properties between cameras. Here, we analyze the changes of features due to image transformations applied to samples from CUHK03-NP, using eight low-level transformation functions with four levels of transformation strength as shown in Figure 10. The feature of the $i$-th image and the feature of its transformed image at level $k$ are denoted by $\boldsymbol{f}_i^{(0)}$ and $\boldsymbol{f}_i^{(k)}$, respectively. For example, for a blurring function, $\boldsymbol{f}_i^{(4)}$ denotes the feature when the $i$-th image is most strongly blurred. Then, we compute the average cosine similarity between displacement vectors of the features after applying a transformation to the images for each level $k$, which is given by $\mathbb{E}_{i,j}[Sim(\boldsymbol{f}_i^{(k)} - \boldsymbol{f}_i^{(k-1)}, \boldsymbol{f}_j^{(k)} - \boldsymbol{f}_j^{(k-1)})]$. The result is shown in Figure 3(a). We observe that, for certain transformations such as decreasing brightness, the displacement vector ($\boldsymbol{f}_i^{(k)} - \boldsymbol{f}_i^{(k-1)}$) due to the transformation is similar across different images to some extent , which is analogous to the effect of camera variations.

**Normalization for image properties**   Then, can we reduce biases of the model towards the low-level properties by utilizing the feature normalization? To find out, we calculate the brightness, sharpness, contrast, and area of all samples, as visualized in Figure 11. Note that all samples in the dataset have almost same contrast values. We divide the samples into $N$ groups of equal size for each property. For example, when dividing the samples into $N = 2$ groups based on the brightness, we use the median brightness value as the threshold for group assignment. Here, a small but meaningful correlation between these group labels and camera labels is observed as shown in Figure 12. Then, we perform a group-specific feature normalization on the features using the property group labels. As presented in Figure 3(b), the normalization on the features based on the property groups is effective for brightness, sharpness, and area. It does not work for contrast since all contrast values are almost equal. In addition, we subdivide each property group into multiple groups based on the camera labels

Table 2: Feature normalization for body angle. "All" includes images of front, back, and side angles.

| Normalization | All | | Front-only | | Side-only | |
|---|---|---|---|---|---|---|
| | mAP | R1 | mAP | R1 | mAP | R1 |
| None | 68.3 | 76.0 | 84.1 | 84.8 | 84.3 | 83.3 |
| Angle-specific | 75.0 | 81.1 | - | - | - | - |
| Camera-specific | 76.2 | 84.1 | **91.1** | **91.5** | **89.4** | **91.3** |
| (Angle, Camera)-specific | **80.5** | **87.2** | - | - | - | - |

and present the result of group-specific normalization with the subdivided group labels in Figure 3(c). Interestingly, the (property group, camera)-specific normalization outperforms the camera-specific normalization for proper $N$ values. For example, compared to the camera-specific normalization, (area group, camera)-specific normalization exhibits about 1.5 mAP improvements. This implies that further considerations of other bias types of ReID models along with the camera bias are needed. Experimental details and additional results are provided in Appendix C.

**Normalization for body angle**  It has been shown that ReID models have a bias towards the body angle of an image (Sun & Zheng, 2019). This bias is likely to be closely related with the camera bias, since the distribution of body angles would be different for each camera orientation. Then, can we reduce the bias of the model towards the body angle by using the feature normalization? To find out, we define three body angle classes (front, back, and side) and construct four angle-labeled datasets from Market-1501, including front-only, side-only, back-only, and all-angle dataset. Then, we perform an angle-specific feature normalization on the features using the angle labels, as well as (angle, camera)-specific normalization like previous paragraph. As shown in Table 2, the angle-specific normalization works and the performance is further improved by using the camera labels together. Details of the datasets are described in Appendix D.

In summary, we observed consistent feature movements due to image transformations and confirmed the applicability of the feature normalization to detailed bias factors such as low-level image properties and body angles. It is encouraging that considering both camera labels and other factors in normalization can achieve performance beyond only considering the camera labels, highlighting the need for further research into biases of ReID models beyond the camera bias. The normalization methods could serve as an easy tool for such exploration.

## 4.4  MORE EMPIRICAL RESULTS

**Generalizability**  We present the evaluation results of the camera-specific feature normalization on multiple ReID models in Table 3. The mean average precision (mAP) and cumulative matching characteristics (CMC) Rank-1 (R1) are used for evaluation. The NMI scores of clustering results are also reported as in Section 3. Note that ISR and PAT (Ni et al., 2023) are domain generalized methods. The feature normalization significantly improves the performance of all models on the unseen domain (white background in the table), regardless of training methods or backbones architectures. For example, on Market-1501, CC (Dai et al., 2022) exhibits about 7.5% improvement in mAP and about 5.9% reduction in camera bias, and TransReID (He et al., 2021) shows about 9.4% improvement in mAP and about 2.7% reduction in camera bias. For the seen domain, the camera-agnostic unsupervised models show slight improvement (gray background), while the camera-aware or supervised models exhibit no improvement (red background). It is likely because the camera bias of these models is already relatively small on the seen domain, *e.g.*, SOLIDER (Chen et al., 2023) has an almost identical bias value to the ground truth. The normalization results of Figure 1(b) are shown in Figure 4, where the less separable distributions are observed. The feature visualization result is illustrated in Figure 15.

**Ablation study**  The camera-specific feature normalization consists of (1) camera-specific, (2) mean centering, and (3) scaling by standard deviation on the features. We investigate the effectiveness of each component for CUHK03-NP in Table 4, using TransReID-SSL (Luo et al., 2021b) trained on MSMT17. ZCA whitening is also evaluated to check the effectiveness of rotation related to covariance across feature dimensions. There are some gains from the entire transforms, but the camera-specific transforms outperform them. It is observed that the camera-specific mean centering has a dominant effect and the scaling operation provides a small but additional gain. The rotation by the ZCA whitening does not exhibit definite gains compared to the normalization.

Table 3: Evaluation results of the camera-specific feature normalization for various ReID models. The numbers denote the performance before/after normalization. "SL" and "CA" denote the supervised learning and camera-aware methods, respectively. "†" indicates transformer backbones. "*" indicates our reproduced results. ISR is trained on external videos and the others are trained on MSMT17.

(a) ReID performance

| Method | SL | CA | Market-1501 mAP | Market-1501 R1 | MSMT17 mAP | MSMT17 R1 | CUHK03-NP mAP | CUHK03-NP R1 | PersonX mAP | PersonX R1 |
|---|---|---|---|---|---|---|---|---|---|---|
| SPCL (Ge et al., 2020) | ✗ | ✗ | 16.0 / 21.9 | 37.1 / 43.7 | 19.1 / 20.3 | 42.4 / 44.4 | 6.1 / 9.1 | 5.1 / 8.1 | 20.4 / 31.0 | 41.2 / 53.7 |
| CC* (Dai et al., 2022) | ✗ | ✗ | 22.5 / 30.0 | 47.3 / 56.4 | 29.8 / 32.2 | 57.1 / 60.4 | 8.4 / 13.5 | 8.3 / 13.1 | 24.7 / 36.8 | 51.4 / 62.1 |
| PPLR (Cho et al., 2022) | ✗ | ✗ | 25.2 / 31.8 | 53.7 / 61.6 | 30.6 / 31.7 | 59.5 / 62.4 | 10.1 / 14.2 | 9.4 / 13.5 | 30.7 / 39.4 | 57.3 / 68.2 |
| TransReID-SSL† (Luo et al., 2021b) | ✗ | ✗ | 53.6 / 62.3 | 78.1 / 83.5 | 49.5 / 53.0 | 75.0 / 77.3 | 27.2 / 35.7 | 25.4 / 34.4 | 45.4 / 59.6 | 65.7 / 79.0 |
| ISR† (Dou et al., 2023) | ✗ | ✗ | 70.2 / 71.9 | 87.0 / 87.8 | 32.5 / 34.2 | 58.8 / 60.8 | 38.6 / 42.3 | 37.1 / 40.5 | 66.4 / 70.2 | 83.1 / 85.3 |
| CAP* (Wang et al., 2021) | ✗ | ✓ | 30.8 / 36.6 | 58.9 / 65.3 | 36.3 / 36.6 | 67.5 / 67.7 | 15.5 / 17.9 | 16.3 / 18.7 | 36.9 / 45.0 | 64.6 / 72.7 |
| ICE-CAM* (Chen et al., 2021) | ✗ | ✓ | 25.9 / 34.9 | 53.4 / 63.3 | 37.8 / 37.9 | 66.9 / 67.5 | 12.2 / 17.2 | 11.9 / 16.4 | 26.2 / 39.8 | 52.2 / 66.7 |
| PPLR-CAM (Cho et al., 2022) | ✗ | ✓ | 28.4 / 34.5 | 58.3 / 65.1 | 42.2 / 41.3 | 73.2 / 73.3 | 12.0 / 16.2 | 12.0 / 15.9 | 31.0 / 39.6 | 57.4 / 68.6 |
| CAJ (Chen et al., 2024) | ✗ | ✓ | 30.6 / 36.9 | 61.3 / 68.1 | 44.3 / 42.8 | 75.1 / 74.4 | 14.1 / 18.1 | 14.6 / 19.1 | 32.5 / 40.9 | 59.5 / 70.0 |
| PAT†* (Ni et al., 2023) | ✓ | ✗ | 43.8 / 52.9 | 70.4 / 76.8 | 54.8 / 54.1 | 78.0 / 78.3 | 24.5 / 29.8 | 24.2 / 29.9 | 50.0 / 59.8 | 72.8 / 80.4 |
| SOLIDER† (Chen et al., 2023) | ✓ | ✗ | 72.4 / 79.2 | 89.0 / 91.7 | 77.1 / 77.0 | 90.7 / 90.6 | 53.9 / 58.7 | 53.8 / 59.1 | 55.4 / 63.4 | 79.5 / 84.8 |
| TransReID† (He et al., 2021) | ✓ | ✓ | 43.1 / 52.5 | 69.5 / 76.1 | 67.8 / 66.7 | 85.4 / 85.0 | 29.9 / 34.5 | 28.8 / 34.7 | 57.7 / 65.8 | 76.9 / 82.8 |

(b) Clustering result

| Method | SL | CA | Market-1501 Bias | Market-1501 Accuracy | MSMT17 Bias | MSMT17 Accuracy | CUHK03-NP Bias | CUHK03-NP Accuracy | PersonX Bias | PersonX Accuracy |
|---|---|---|---|---|---|---|---|---|---|---|
| SPCL (Ge et al., 2020) | ✗ | ✗ | 22.5 / 15.1 | 75.0 / 79.5 | 34.7 / 32.2 | 84.0 / 84.5 | 18.2 / 14.7 | 71.2 / 74.0 | 22.0 / 13.2 | 74.3 / 80.6 |
| CC* (Dai et al., 2022) | ✗ | ✗ | 17.1 / 11.2 | 81.0 / 84.7 | 32.5 / 29.7 | 88.0 / 89.0 | 17.6 / 10.8 | 74.6 / 78.4 | 20.6 / 9.5 | 78.9 / 85.1 |
| PPLR (Cho et al., 2022) | ✗ | ✗ | 15.6 / 9.9 | 81.7 / 85.2 | 30.2 / 27.0 | 89.0 / 89.9 | 15.9 / 10.1 | 77.4 / 80.5 | 15.3 / 6.3 | 82.0 / 87.5 |
| TransReID-SSL† (Luo et al., 2021b) | ✗ | ✗ | 9.7 / 7.2 | 92.2 / 94.3 | 27.1 / 25.4 | 92.8 / 93.6 | 7.0 / 4.0 | 84.2 / 86.9 | 12.5 / 3.9 | 88.8 / 93.5 |
| ISR† (Dou et al., 2023) | ✗ | ✗ | 9.7 / 9.4 | 95.8 / 96.0 | 30.3 / 29.6 | 89.4 / 89.9 | 5.4 / 4.5 | 87.7 / 88.9 | 6.1 / 4.5 | 94.9 / 95.9 |
| CAP* (Wang et al., 2021) | ✗ | ✓ | 12.3 / 8.2 | 84.0 / 86.9 | 25.5 / 24.8 | 90.1 / 90.1 | 8.0 / 6.5 | 78.8 / 80.6 | 9.9 / 5.2 | 86.0 / 89.6 |
| ICE-CAM* (Chen et al., 2021) | ✗ | ✓ | 13.9 / 9.1 | 79.2 / 82.9 | 26.8 / 25.4 | 90.6 / 90.8 | 12.7 / 7.3 | 77.5 / 80.7 | 18.1 / 7.4 | 78.3 / 86.1 |
| PPLR-CAM (Cho et al., 2022) | ✗ | ✓ | 14.3 / 9.7 | 84.1 / 87.2 | 26.7 / 25.5 | 92.4 / 92.5 | 13.7 / 10.0 | 78.4 / 82.6 | 14.6 / 6.4 | 81.8 / 88.0 |
| CAJ (Chen et al., 2024) | ✗ | ✓ | 12.2 / 8.9 | 82.1 / 84.2 | 25.2 / 24.3 | 92.7 / 92.4 | 11.6 / 8.3 | 80.8 / 83.3 | 12.2 / 5.7 | 83.3 / 88.0 |
| PAT†* (Ni et al., 2023) | ✓ | ✗ | 10.8 / 9.1 | 90.8 / 93.3 | 24.0 / 23.4 | 94.3 / 94.6 | 6.4 / 4.4 | 85.3 / 86.5 | 7.7 / 3.5 | 90.8 / 93.9 |
| SOLIDER† (Chen et al., 2023) | ✓ | ✗ | 7.3 / 6.9 | 96.5 / 97.5 | 21.3 / 21.4 | 96.9 / 96.9 | 1.6 / 1.6 | 90.8 / 92.8 | 2.8 / 1.5 | 93.8 / 95.1 |
| TransReID† (He et al., 2021) | ✓ | ✓ | 13.6 / 10.9 | 89.8 / 92.2 | 23.6 / 22.6 | 94.5 / 95.2 | 3.9 / 3.6 | 84.7 / 86.8 | 6.6 / 3.3 | 92.7 / 95.1 |
| Ground Truth | - | - | 6.4 | - | 19.2 | - | 0.1 | - | 0.0 | - |

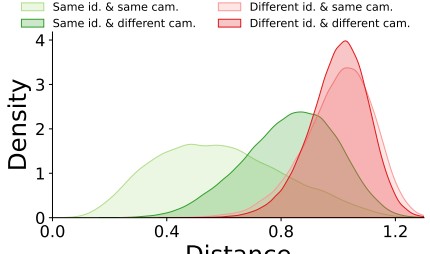

Figure 4: Normalization result of Figure 1(b).

Table 4: Ablation study of feature normalization.

| Method | Entire mAP | Entire R1 | Camera-specific mAP | Camera-specific R1 |
|---|---|---|---|---|
| Baseline | 27.2 | 25.4 | 27.2 | 25.4 |
| + Centering | 27.8 | 25.8 | 34.7 | 33.0 |
| + Scaling | 27.7 | 25.6 | 27.2 | 25.1 |
| + Centering + Scaling | 28.8 | 26.6 | 35.7 | 34.4 |
| + ZCA whitening | 18.7 | 19.1 | 30.1 | 35.1 |

**Image volume for computing normalization parameters**  We explore the impact of the number of samples used to compute the statistics for normalization in Figure 5. Equal numbers of samples are randomly sampled from each camera in Market-1501. TransReID-SSL trained on MSMT17 is used in this experiments. The performance is degraded when using too few samples per camera (*e.g.*, five samples). Interestingly, using more than 25 samples per camera leads to positive effects and the gains start to be saturated over 100 samples. This suggests that the number of samples needed to represent camera features is not very large.

**Combination with other postprocessing methods**  We test whether the camera-specific feature normalization is still effective when used in conjunction with other postprocessing algorithms, using TransReID-SSL trained on MSMT17. As shown in Table 5, the normalization brings about 9% gains in mAP for all methods. In other words, the problem of camera bias remains after applying the conventional postprocessing methods. Note that unlike other methods resulting in decreases in R5 and R10, the normalization consistently improves all metrics.

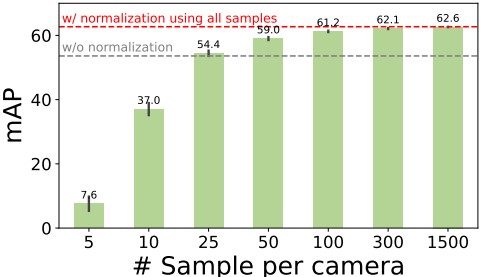

Figure 5: Results based on the number of samples used to calculate normalization parameters.

Table 5: Combination results with other feature postprocessing methods on Market-1501.

| Postprocessing | mAP | R1 | R5 | R10 |
|---|---|---|---|---|
| None | 53.6 | 78.1 | 89.2 | 92.3 |
| + Normalization | **62.3** | **83.5** | **92.8** | **95.5** |
| DBA (Gordo et al., 2017) | 59.2 | 77.9 | 88.4 | 91.8 |
| + Normalization | **68.6** | **84.2** | **92.7** | **95.1** |
| AQE (Chum et al., 2007) | 61.1 | 78.9 | 87.5 | 90.7 |
| + Normalization | **70.6** | **85.1** | **91.8** | **94.3** |
| Reranking (Zhong et al., 2017b) | 67.7 | 78.4 | 85.2 | 88.6 |
| + Normalization | **76.3** | **84.7** | **90.7** | **93.0** |

## 5 RISK OF CAMERA BIAS IN UNSUPERVISED LEARNING

### 5.1 RISK OF BIASED PSEUDO LABELS

In Section 3, we observed that the ReID models learned in unsupervised manners have a large camera bias even on their training data. We argue that the existing USL algorithms have two limitations introducing the camera bias into the models. First, the pseudo labels of training data are biased towards the camera labels. In USL, a model is supervised by the pseudo labels of the training samples which are usually generated by clustering of the features extracted by the model. However, as we have seen, the clustering result is already camera-biased, hence using them for training would make the model dependent on the camera-related information. Second, camera-biased clusters with few cameras are used in training without sufficient consideration. For example, consider a cluster only consisting of samples from one camera. Since most of the samples of this cluster may share similar camera-related information (*e.g.*, background), utilizing them as positive training samples can lead the model to pay more attention to the common camera-related information. Also, the samples in that cluster were likely grouped together incorrectly due to the shared camera information, which is expected to be more common in the early stage of model training.

### 5.2 TOY EXAMPLE RESULTS

We investigate the risks of biased training data toward cameras using toy examples. ResNet50 models are trained on the toy datasets using the cluster contrastive loss (Dai et al., 2022) in the experiments.

Figure 6(a) compares the training results with the different levels of camera bias and accuracy of pseudo labels for the same training samples. We constructed a dataset of 7500 samples by randomly selecting 500 identities from Market-1501, where each identity has 5 samples per camera with 3 cameras. To generate pseudo labels with varying degrees of camera bias at the same accuracy, a certain proportion of identities were divided into three equally-sized clusters for each identity based either on camera labels ("Camera") or random selection ("Random"). Five splitting ratios of 0%, 25%, 50%, 75%, and 100% were used. For example, the pseudo labels generated by splitting 50% of the identities consist of 250 original clusters and 750 split clusters, totaling 1000 clusters. The bias of the pseudo labels is measured by calculating the mean entropy of the camera labels within each cluster (Lee et al., 2023). It is observed that, at the same pseudo label accuracy, models trained with "Camera" consistently perform worse than those trained with "Random". Moreover, "Random" with 91.9% accuracy outperforms "Camera" with 93.8% accuracy, despite having lower accuracy. These results suggest that greater camera bias of pseudo labels has a detrimental effect on model training, and pseudo labels with lower accuracy but less camera bias can provide more benefits than those with higher accuracy but greater camera bias.

Figure 6(b) illustrates the impact of camera diversity of training data, using ground truth labels. We constructed five datasets of 11821 samples of 1041 identities from MSMT17, where the maximum numbers of cameras per identity are different. As expected, the model performance declines as the maximum number of cameras decreases. Notably, a significant performance drop is observed when the model is trained with samples from only a single camera for each identity. This suggests that using single-camera clusters for training can degrade the model performance. In addition, the influence of clustering parameter on the camera bias is investigated in Appendix I.

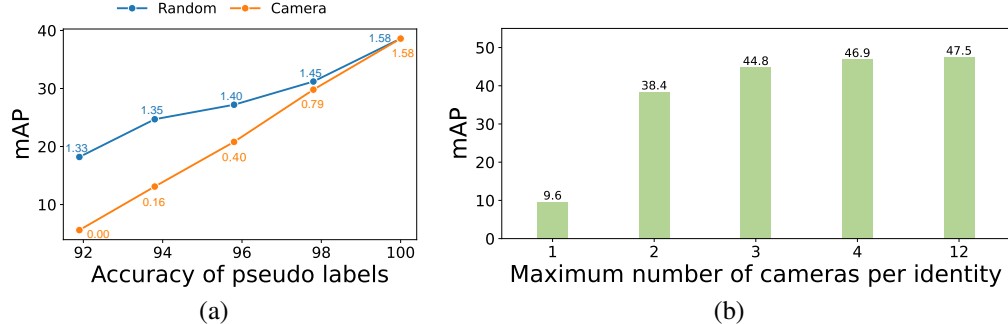

Figure 6: Risk of biased clusters. (a) Training results with varying pseudo label qualities for the same training samples. Two pseudo label generation methods, "Random" and "Camera", are used. The colored numbers denote the average camera entropy of the clusters. (b) Training results with varying maximum number of cameras per identity for the same number of samples.

---

**Algorithm 1** Unsupervised learning algorithm for Person ReID with simple modificaitons

---
**Require:** Initialized backbone encoder $f_\theta$ and training samples with camera labels $\mathcal{X}$
  **for** n in $[1, \text{num\_epochs}]$ **do**
    Extract features $\mathcal{F}$ from $\mathcal{X}$ by $f_\theta$.
    **(1) Debiased pseudo labeling:**
      Transform $\mathcal{F}$ to $\hat{\mathcal{F}}$ by applying the camera-specific feature normalization.
      Generate pseudo labels by clustering $\hat{\mathcal{F}}$.
    **(2) Discarding biased clusters:**
      Collect the images belong to the clusters of single camera as $\mathcal{B}$.
      Reconstruct training images by $\mathcal{X}' = \mathcal{X} - \mathcal{B}$.
    Prepare for training iterations (*e.g.*, initialization of feature memory).
    **for** i in $[1, \text{num\_iterations}]$ **do**
      Sample a mini-batch from the reconstructed data $\mathcal{X}'$.
      Compute loss (*e.g.*, contrastive loss).
      Update the encoder $f_\theta$.
      Update auxiliary modules (*e.g.*, update of feature memory).
    **end for**
  **end for**

---

## 5.3 SIMPLE STRATEGIES FOR DEBIASED UNSUPERVISED LEARNING

To reduce the explored risk of camera bias in unsupervised learning, we present two simple training strategies applicable to existing USL algorithms; **(1) debiased pseudo labeling**: clustering on the debiased features computed by Equation 2 instead of the original features when generating pseudo labels, and **(2) discarding biased clusters**: discarding the single-camera clusters in training data. We present an example of applying the proposed strategies to unsupervised learning in Algorithm 1. With these minor modifications, we observe significant performance improvements in next section.

## 5.4 EMPIRICAL RESULTS

We validate the suggested training strategies on the SOTA camera-agnostic methods, CC and PPLR, and the SOTA camera-aware method, PPLR-CAM. A vehicle ReID dataset, VeRi-776 (Liu et al., 2016), is additionally used and the person and vehicle images are resized to $384 \times 128$ and $256 \times 256$, respectively, following the setup of PPLR. The models are trained on a H100 GPU with batch size 256 and 100 training epochs, with DBSCAN (Ester et al., 1996) to obtain pseudo labels. Our strategies effectively improves all methods as presented in Table 6. In particular, outstanding performance gains are obtained on the challenging benchmark, MSMT17, *e.g.*, 19.3% mAP increase for CC. The gains for PPLR-CAM are relatively small, which is likely because it uses a camera-aware loss function. In addition, the number of discarded training samples by our strategy is discussed in Appendix H.

**Ablation study** Table(a) of Figure 7 investigates the individual effect of our strategies, using CC. Both of the suggested strategies contribute to the performance improvements by reducing the camera bias. The ratio of the single-camera clusters and clustering accuracy during training are illustrated in the plot of Figure 7. It is observed that the baseline has unusually high single-camera cluster rates (from about 80% to about 35%), which is effectively mitigated by the proposed methods.

Table 6: Results of modifying the existing USL algorithms based on our training strategies. "*" indicates our reproduced result with the official code.

| Method | Market-1501 | | | | MSMT17 | | | | VeRi-776 | | | |
|---|---|---|---|---|---|---|---|---|---|---|---|---|
| | mAP | R1 | R5 | R10 | mAP | R1 | R5 | R10 | mAP | R1 | R5 | R10 |
| *Camera-agnostic unsupervised* | | | | | | | | | | | | |
| SPCL (Ge et al., 2020) | 73.1 | 88.1 | 95.1 | 97.0 | 19.1 | 42.3 | 55.6 | 61.2 | 36.9 | 79.9 | 86.8 | 89.9 |
| ICE (Chen et al., 2021) | 79.5 | 92.0 | 97.0 | 98.1 | 29.8 | 59.0 | 71.7 | 77.0 | - | - | - | - |
| PPLR (Cho et al., 2022) | 81.5 | 92.8 | 97.1 | 98.1 | 31.4 | 61.1 | 73.4 | 77.8 | 41.6 | 85.6 | 91.1 | 93.4 |
| CC (Dai et al., 2022) | 83.0 | 92.9 | 97.2 | 98.0 | 33.0 | 62.0 | 71.8 | 76.7 | 40.8 | 86.2 | 90.5 | 92.8 |
| *Camera-aware unsupervised* | | | | | | | | | | | | |
| CAP (Wang et al., 2021) | 79.2 | 91.4 | 96.3 | 97.7 | 36.9 | 67.4 | 78.0 | 81.4 | - | - | - | - |
| ICE-CAM (Chen et al., 2021) | 82.3 | 93.8 | 97.6 | 98.4 | 38.9 | 70.2 | 80.5 | 84.4 | - | - | - | - |
| PPLR-CAM (Cho et al., 2022) | 84.4 | 94.3 | 97.8 | 98.6 | 42.2 | 73.3 | 83.5 | 86.5 | 43.5 | 88.3 | 92.7 | 94.4 |
| PPLR* (Cho et al., 2022) | 77.4 | 89.6 | 96.1 | 97.4 | 27.2 | 55.7 | 67.1 | 71.8 | 41.5 | 85.6 | 91.4 | 93.2 |
| PPLR* (Cho et al., 2022) + Ours | **84.6** | **93.9** | **97.8** | **98.6** | **40.7** | **71.4** | **82.3** | **85.4** | **43.2** | **86.7** | **91.7** | **93.7** |
| PPLR-CAM* (Cho et al., 2022) | 84.1 | **94.0** | 97.7 | 98.6 | 40.7 | 71.8 | 82.6 | 85.7 | 43.3 | 88.1 | 92.2 | 94.2 |
| PPLR-CAM* (Cho et al., 2022) + Ours | **84.3** | 93.8 | **98.1** | **98.8** | **44.4** | **75.8** | **84.9** | **87.7** | **43.7** | **88.2** | **92.8** | **94.5** |
| CC* (Dai et al., 2022) | 82.6 | 91.8 | 96.7 | 97.8 | 29.8 | 57.1 | 68.5 | 72.8 | 38.2 | 79.8 | 83.9 | 86.9 |
| CC* (Dai et al., 2022) + Ours | **85.2** | **93.5** | **97.3** | **98.2** | **49.1** | **76.5** | **85.6** | **88.3** | **45.3** | **89.8** | **93.9** | **95.3** |

| Method | mAP | R1 | R5 | R10 | Bias |
|---|---|---|---|---|---|
| Baseline | 29.8 | 57.1 | 68.5 | 72.8 | 32.5 |
| (a) Ablation study | | | | | |
| + (1) Debiased pseudo labeling | 44.6 | 71.9 | 82.0 | 85.1 | 26.5 |
| + (2) Discarding biased clusters | 45.4 | 73.7 | 83.4 | 86.6 | 25.4 |
| + Both of (1) and (2) | **49.1** | **76.5** | **85.6** | **88.3** | **23.8** |
| (b) Cluster-weighted loss | | | | | |
| + CD loss (Lee et al., 2023) | 40.7 | 69.4 | 80.2 | 83.8 | 25.1 |
| + Binary weighting | 40.1 | 69.3 | 79.9 | 83.3 | 25.9 |
| Ground Truth | - | - | - | - | 19.2 |

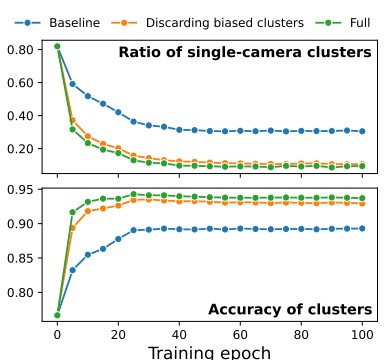

Figure 7: Results of our training strategies for debiased unsupervised learning on MSMT17.

**Comparison to weighted loss** We experiment with weighted loss methods related to discarding biased clusters in Table(b) in Figure 7. The camera diversity (CD) loss (Lee et al., 2023) weights a sample proportionally to the diversity of cameras within the cluster it belongs to, with 0-weight for single-camera clusters. The binary weighting is a variant of it, where we assign 1-weight to clusters of more than one camera. It is observed that the performance improvement of the CD loss is dominated by the 0-weight setting for single-camera clusters. In our opinion, the CD loss has a side effect of randomizing the mini-batch size or learning rate, since the effective number of samples involved in a model update is randomly changed at each training iteration. Discarding the biased clusters in training is a simpler method free from such drawbacks, outperforming the CD loss.

## 6 CONCLUSION

We revisited the debiasing effects of normalization on embedding vectors of ReID models and explored the risk of camera bias inherent in unsupervised learning for ReID models. We found that the existing ReID models are biased towards camera labels on unseen domain, and the unsupervised models even have a large camera bias to their training data. We analyzed why the camera-specific feature normalization has debiasing effects and explored its potential and applicability for ReID tasks in comprehensive empirical studies. It was observed that, for a camera variation, the sensitivity of each feature dimension is quite different and features tend to move consistently in sensitive dimensions. Then, it was shown that the feature normalization is a simple but effective bias elimination method for ReID models in general, including biases towards low-level properties and body angle. Also, we empirically showed the detrimental effects of biased pseudo labels using toy examples and achieved significant performance improvements with simple modifications to the existing unsupervised algorithms. We hope that the insights from this work will serve as an insightful foundation for researching biases of ReID models and developing debiasing techniques for ReID models.

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

## A STATISTICS OF BENCHMARKS

Table 7: Statistics of datasets used in our experiments. In PersonX, each identity has 36 images for each camera.

| Dataset | # identities | # images | # cameras | Scene |
|---------|-------------|----------|-----------|-------|
| CHUK03-NP | 1,467 | 14,097 | 2 | Indoor |
| Market-1501 | 1,501 | 32,668 | 6 | Outdoor |
| MSMT17 | 4,101 | 126,441 | 15 | Indoor/outdoor |
| PersonX | 1,266 | 273,465 | 6 | Synthetic |
| VeRi-776 | 200 | 51,003 | 20 | Outdoor |

## B ADDITIONAL DISCUSSIONS ON FEATURE SPACE

### B.1 EXPERIMENTAL DETAILS

Here, we explain how we calculate the displacement vectors from feature pairs of two different cameras with the same identity in Section 4.2. Suppose a labeled dataset is given. For the $i$-th identity, we denote its $k$-th image in the $j$-th camera by $\boldsymbol{x}_k^{(i,j)}$. We denote the number of images of the $i$-th identity from the $j$-th camera by $N^{(i,j)}$, *i.e.*, the $i$-th identity has $N^{(i,j)}$ images from the $j$-th camera. For a pretrained encoder $f_\theta$, the feature of $\boldsymbol{x}_k^{(i,j)}$ is given by $\boldsymbol{f}_k^{(i,j)} = f_\theta(\boldsymbol{x}_k^{(i,j)})$. Then, we compute an average representation of the $i$-th identity in the $j$-th camera, $\boldsymbol{s}^{(i,j)}$, as follows:

$$\boldsymbol{s}^{(i,j)} = \mathbb{E}_k[\boldsymbol{f}_k^{(i,j)}] = \frac{1}{N^{(i,j)}} \sum_{k=1}^{N^{(i,j)}} \boldsymbol{f}_k^{(i,j)}. \tag{3}$$

The displacement vector of the $i$-th identity between the $j$-th camera and the $l$-th camera, $\boldsymbol{d}_i^{j \to l}$, is given by

$$\boldsymbol{d}_i^{j \to l} = \boldsymbol{s}^{(i,l)} - \boldsymbol{s}^{(i,j)}. \tag{4}$$

The displacement vector $\boldsymbol{d}_i^{j \to l}$ can be thought as the motion of the feature due to the camera change from the $j$-th camera to the $l$-th camera. The average cosine similarity between the displacement vectors is computed by

$$\mathbb{E}_{p,q}\mathbb{E}_{m,n}[Sim(\boldsymbol{d}_m^{p \to q}, \boldsymbol{d}_n^{p \to q})], \tag{5}$$

where $(p, q)$ and $(m, n)$ denote a pair of different cameras and a pair of different identities, respectively, and $Sim$ denotes the cosine similarity function.

### B.2 ADDITIONAL RESULTS

We additionally analyze the feature space of other models, PPLR-CAM and SOLIDER, following the experimental setup of Section 4.2. The results are presented in Figures 8 and 9. The tendencies previously discussed for TransReID-SSL are also observed for these models.

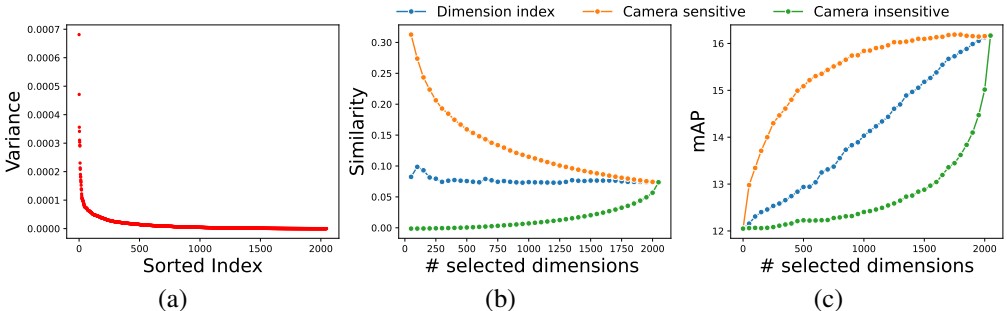

Figure 8: Analysis on the 2048-dimensional embedding space of PPLR-CAM (Cho et al., 2022). (a) Variance of each dimension of camera mean features. (b) Cosine similarity of displacement vectors between feature pairs of the same identity from different cameras for selected dimensions. (c) Result of camera-specific centering on the features for selected dimensions.

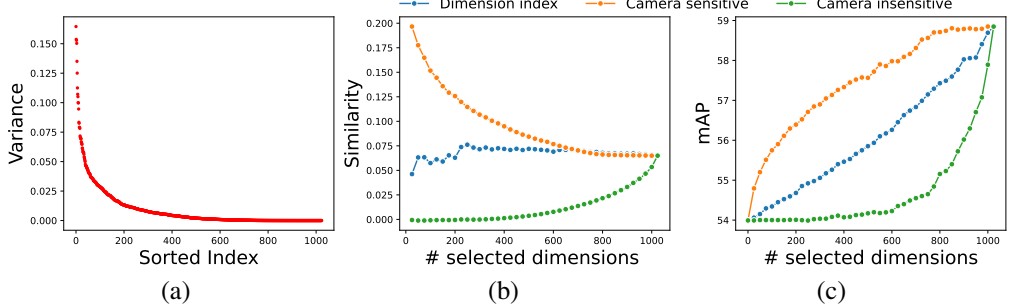

(a)                          (b)                          (c)

Figure 9: Analysis on the 1024-dimensional embedding space of SOLIDER (Chen et al., 2023). (a) Variance of each dimension of camera mean features. (b) Cosine similarity of displacement vectors between feature pairs of the same identity from different cameras for selected dimensions. (c) Result of camera-specific centering on the features for selected dimensions.

# C  ADDITIONAL DISCUSSIONS ON LOW-LEVEL IMAGE PROPERTIES

## C.1  EXPERIMENTAL DETAILS

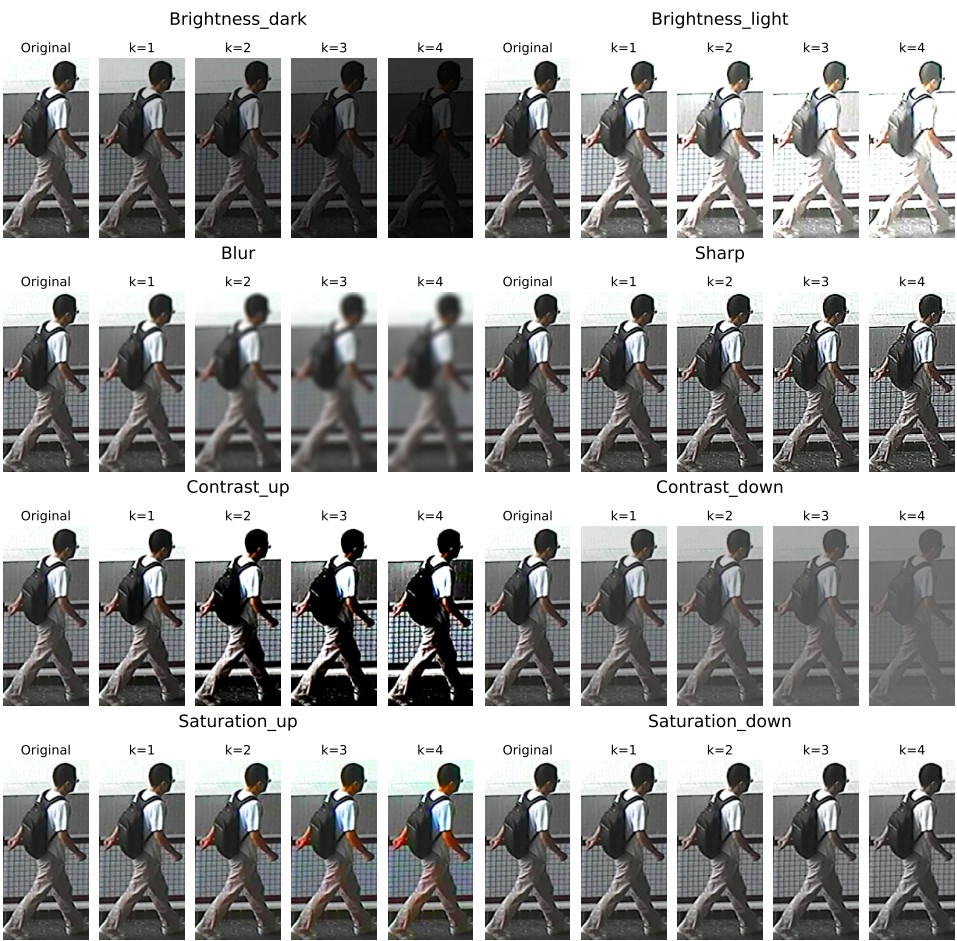

Figure 10: Examples of applying the transformation functions to an image with four strength levels.

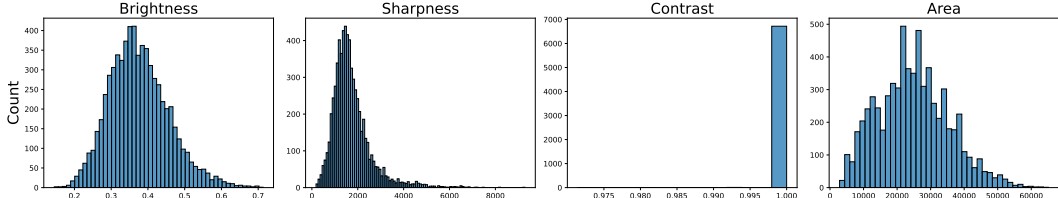

Figure 11: Statistics of low-level properties of images used in our experiments on each low-level property. Note that all images have the same contrast value.

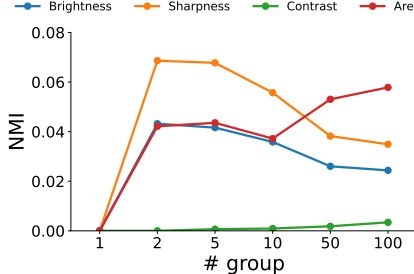

Figure 12: The Normalized Mutual Information (NMI) scores between property group labels and camera labels for each property. A weak correlation between them is observed, except the contrast.

## C.2 ADDITIONAL RESULTS

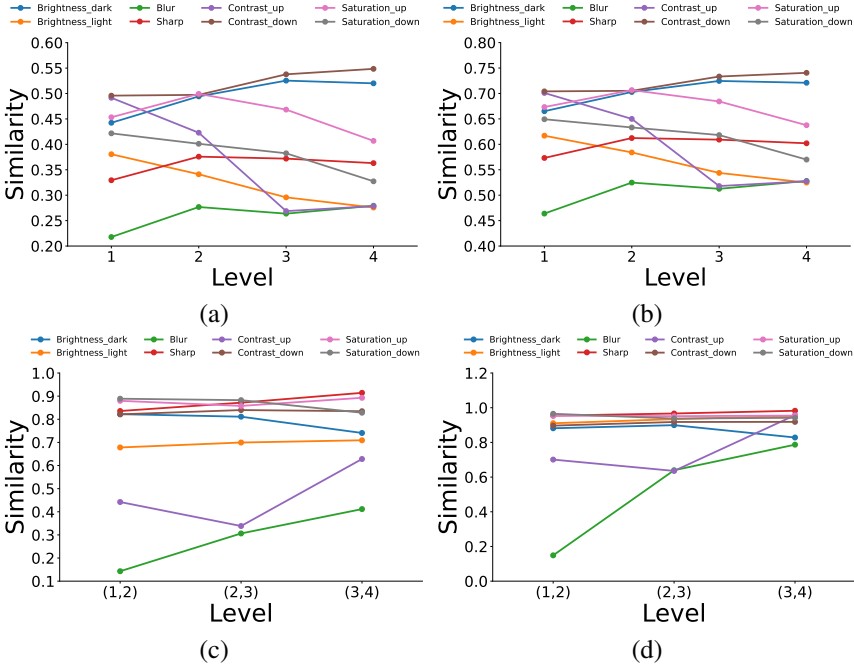

Figure 13: Cosine similarity of displacement vectors of the features due to low-level transformations.

Here, we analyze the movements of the features due to low-level image transformations in multiple aspects. In the experiments, we define four levels of transformation strength for several low-level transformation functions, as shown in Figure 10. For a transformation function, we denote the feature of the $i$-th image and the feature of its transformed image at level $k$ by $\boldsymbol{f}_i^{(0)}$ and $\boldsymbol{f}_i^{(k)}$, respectively.

For example, for the blurring function, $\boldsymbol{f}_i^{(4)}$ denotes the feature when the $i$-th image is most strongly blurred. Then, we denote the displacement vector of the feature after applying the transformation to the images for a level $k$ by $\boldsymbol{d}_i^{(k)} = \boldsymbol{f}_i^{(k)} - \boldsymbol{f}_i^{(k-1)}$. We also denote the average displacement vector in the level $k$ by $\boldsymbol{m}^{(k)} = \mathbb{E}_i[\boldsymbol{d}_i^{(k)}]$. We investigate the tendency of the movements of the features by computing the following cosine similarity between the displacement vectors:

- (a) $\mathbb{E}_{i,j}[Sim(d_i^{(k)}, d_j^{(k)})]$: How similar are the movements of features to each other under a transformation?

- (b) $\mathbb{E}_{i,j}[Sim(d_i^{(k)}, m^{(k)})]$: How similar are the movements of features to the average movement under a transformation?

- (c) $\mathbb{E}_{i,j}[Sim(d_i^{(k)}, d_i^{(k+1)})]$: How similar are the movements of features to their previous motion when a stronger transformation is applied?

- (d) $\mathbb{E}_{i,j}[Sim(m^{(k)}, m^{(k+1)})]$: How consistent is the average movement when a stronger transformation is applied?

The results are presented in Figure 13. It can be observed that the motions of the features due to specific low-level transformations, *e.g.*, "Contrast_down", are similar to each other overall.

## D    EXPERIMENTAL DETAILS ON BODY ANGLE

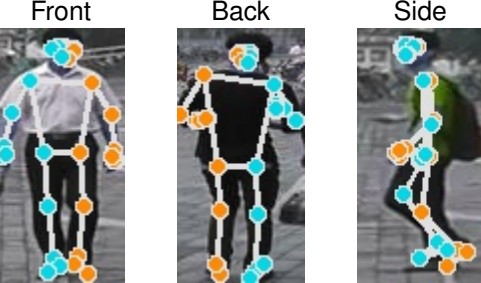

Figure 14: Templates of three body angle classes defined in our experiment. The blue and orange points denote the right and left body parts, respectively.

Table 8: Testset statistics of the experiments on body angle. The number of images in each case is shown.

| Class | CAM 1 | CAM 2 | CAM 3 | CAM 4 | CAM 5 | CAM 6 | All |
|-------|-------|-------|-------|-------|-------|-------|-----|
| Front | 406 | 557 | 639 | 101 | 91 | 243 | 2,037 |
| Back | 410 | 235 | 639 | 121 | 95 | 191 | 1,691 |
| Side | 120 | 63 | 222 | 102 | 44 | 108 | 659 |
| Total | 936 | 855 | 1,500 | 324 | 230 | 542 | 4,387 |

We define three body angle classes of front, back, and side, and construct a test set which is a subset of Market-1501. As shown in Figure 14, we define a template of body keypoints to each class. To obtain the labels of test images, we extract the body keypoints of the images using MediaPipe (Lugaresi et al., 2019), and classify the images through a template-based nearest neighbor classification. The statistics of the constructed dataset are presented in Table 8.

## E    INFLUENCE OF CAMERA BALANCE IN TRAINING DATA

Table 9: Evaluation results of the camera-specific normalization for CC trained on PersonX with the ground truth labels.

| Market-1501 | | MSMT17 | | CUHK03-NP | | PersonX | |
|---|---|---|---|---|---|---|---|
| mAP | R1 | mAP | R1 | mAP | R1 | mAP | R1 |
| 12.7 / **18.8** | 31.6 / **39.3** | 1.2 / **2.3** | 4.0 / **6.6** | 4.5 / **7.0** | 4.4 / **6.5** | 87.8 / **88.8** | 95.4 / **95.9** |

In the widely used training datasets, using Market-1501, MSMT17 and CUHK03-NP, the number of samples of an identity varies for each camera view. In other words, there is an inherent camera imbalance in these datasets, which may induce the camera bias into the model during training. Then, if this imbalance is corrected, would the camera bias be resolved? To find out, CC is trained on PersonX with ground truth labels, where all identities have the same number of samples in each camera. Table 9 presents the evaluation results of the camera-specific normalization on the model for several benchmarks. It is observed that the effect of debiasing is still definite, suggesting that the camera bias still exists even when there is no camera imbalance in the training data. In other words, more effort is required for debiasing beyond balancing the training dataset.

## F    FEATURE VISUALIZATION RESULT

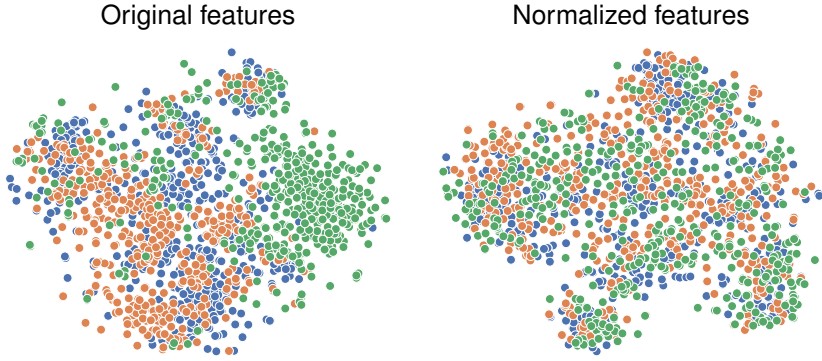

Figure 15: The t-SNE result of features of PPLR-CAM trained on MSMT17 using samples from Market-1501. Different colors are used for each camera.

Figure 15 presents the t-SNE result of features of PPLR-CAM trained on the MSMT dataset using samples from the Market dataset. It is observed that the features from the same camera tend to cluster more than the features from the different cameras in the left plot. This camera bias is effectively mitigated by the normalization as shown in the right plot.

## G    ADDITIONAL RESULT OF CAMERA-SPECIFIC NORMALIZATION

Table 10: Evaluation result of SPCL trained in an unsupervised domain adaptive manner, with Market-1501 and MSMT17 as the source domain and target domain, respectively. The numbers denote the performance before/after the camera-specific normalization.

| Market-1501 | | MSMT17 | | CUHK03-NP | | PersonX | |
|---|---|---|---|---|---|---|---|
| mAP | R1 | mAP | R1 | mAP | R1 | mAP | R1 |
| **86.8** / 86.1 | **94.7** / 93.9 | 26.8 / **28.5** | 53.7 / **56.1** | 13.9 / **18.9** | 13.3 / **18.1** | 36.1 / **45.4** | 59.2 / **68.9** |

## H    NUMBER OF DISCARDED TRAINING SAMPLES BY OUR USL STRATEGY

Table 11: The proportion of discarded training samples by our training strategy.

| Epoch | 0 | 20 | 40 | 60 | 80 | 100 |
|---|---|---|---|---|---|---|
| Discarded samples | 63.5% | 5.8% | 3.0% | 2.6% | 2.8% | 2.8% |

Discarding biased clusters in Section 5.3 reduces the effective number of training samples. Table 11 presents the ratio of the discarded samples (*i.e.*, the samples of single-camera clusters) during training of CC with our training strategy. We observe a drastic discarding ratio in the initial epoch of model training. However, the proportion rapidly reduces; only approximately 3% of the total samples are excluded in the last epochs. As a result, the risk is reduced by excluding many samples in the early training stages, and as the model converges, it learns enough knowledge from almost all samples. Note that, the suggested learning strategy is an effective and easy-to-implement solution, which significantly improves the mAP of this model by 19.3%.

## I    INFLUENCE OF CLUSTERING PARAMETER IN USL

Table 12: Training results of CC with the varying $\epsilon$ parameter of the DBSCAN algorithm.

| Method | mAP | R1 | R5 | R10 | Bias | # training clusters |
|---|---|---|---|---|---|---|
| (a) Without our training strategies | | | | | | |
| $\epsilon = 0.4$ | 18.3 | 39.2 | 49.7 | 54.8 | 33.7 | 2495 |
| $\epsilon = 0.5$ | 21.8 | 45.8 | 56.5 | 61.3 | 32.0 | 2090 |
| $\epsilon = 0.6$ | 29.8 | 57.1 | 68.5 | 72.8 | 32.5 | 1564 |
| $\epsilon = 0.7$ | 32.0 | 58.6 | 71.1 | 75.9 | 30.5 | 1108 |
| $\epsilon = 0.8$ | 8.2 | 19.1 | 27.9 | 32.9 | 40.7 | 291 |
| (b) With our training strategies | | | | | | |
| $\epsilon = 0.4$ | 44.5 | 74.3 | 84.0 | 86.6 | 24.6 | 1708 |
| $\epsilon = 0.5$ | 46.9 | 75.1 | 84.6 | 87.2 | 24.4 | 1516 |
| $\epsilon = 0.6$ | 49.1 | 76.5 | 85.6 | 88.3 | 23.8 | 1245 |
| $\epsilon = 0.7$ | 46.2 | 73.5 | 83.5 | 86.8 | 24.0 | 974 |
| $\epsilon = 0.8$ | - | - | - | - | - | - |

Since the clustering result depends on the parameter settings of the clustering algorithm, it is possible that the camera bias of the model varies as well. Here, we investigate the influence of the most important parameter of DBSCAN, $\epsilon$, which is the maximum distance between two samples to be neighborhood. The training results of CC on MSMT using several $\epsilon$ values are presented in Table 12, where the number of clusters at the last training epoch of each model is also shown.

In Table 12(a), it is observed that the larger $\epsilon$ values tend to roughly decrease the camera bias, while the too large value ($\epsilon = 0.8$) results in the severe performance degradation. A larger $\epsilon$ value can make it easier for samples to cluster together, leading to fewer clusters and larger cluster sizes. Thus, it seems that as the $\epsilon$ value increases, the diversity of cameras in each cluster benefits from the increased cluster size up to a certain point. However, the samples are indiscriminately clustered for a too large value, causing poor clustering quality. In this context, setting an appropriate value for $\epsilon$ is crucial for model training. We find that $\epsilon = 0.7$ yields the best result, with the number of clusters (1108) most similar to the number of identities (1041) in the training data. In Table 12(b), it is shown that the performance of the models are considerably improved by our training strategies. The result of $\epsilon = 0.8$ is omitted since the number of the generated clusters was extremely low, so the mini-batches were not produced properly with the same data sampler used in the previous experiments.

## J ADDITIONAL DISCUSSIONS

**Limitations**    The camera-specific feature normalization requires additional computation of mean and variance for each camera, followed by the normalization on the features. The expected running time of these operations increases linearly with the number of data. The calculation of the statistics can be a memory-exhaustive process if the number of samples per a camera is large. To solve the problem of computational cost, for example, general under-sampling techniques can be adopted. This issue is left as a topic for future work.

**Broader impacts**    Our work focuses on the re-identification technology, which is widely used in real-world applications such as surveillance systems and traffic management solutions. By mitigating bias in target camera domains with a simple approach at the inference level, deploying these applications becomes easier, leading to broader use of AI-powered solutions. As a negative societal impact of the work, an improvement in the re-identification could be used to surveil people in negative manners.

**Reproducibility**    The code for reproducing the experiment results is provided in the supplementary material.

