# OpenReview forum: "Exploring the Camera Bias of Person Re-identification"
_ICLR.cc/2025/Conference — ICLR 2025 Spotlight_

### Official Review · Reviewer_ffre · 2024-10-26

**Soundness:** 2
**Presentation:** 3
**Contribution:** 2
**Rating:** 6
**Confidence:** 4

**Summary:**

To tackle the camera bias issue in person ReID models, this paper offers a comprehensive analysis and proposes a straightforward yet effective test-time post-processing method. Experiments conducted on several ReID datasets demonstrate the method's effectiveness. Overall, the paper is well-written and thorough, but the comparison experiments are somewhat limited, and the novelty of the proposed method may be restricted.

**Strengths:**

1.The paper is thorough and well-written.

2.The paper offers a comprehensive analysis to address the camera bias issue in person ReID models.

**Weaknesses:**

1.As similar feature normalization techniques [R1, R2] have been extensively explored in domain generalization (DG) and domain adaptation (DA) tasks, my main concern is that the novelty of this paper may be limited. It would be beneficial to highlight the advantages of this paper compared to existing DG methods for clearer presentation.

2.In my opinion, the effect of the proposed method is similar to that of domain generalization (DG) methods. Therefore, it is not appropriate to directly compare it with supervised or unsupervised methods in the DG setting (training on MSMT and testing on other datasets). For a more thorough comparison, it would be helpful to include additional related DG and DA methods in the experiments.

3.In Table 3, it shows that the improvement achieved by combining the proposed method with the supervised person ReID task (highlighted in red) is limited. More analysis is needed to explain the reasons for this limitation in this context.

4.Since ICE-CAM and PPLR-CAM have demonstrated that unsupervised person ReID methods can benefit from camera information, the improvements observed in the unsupervised ReID setting (highlighted in gray) may be due to the incorporation of camera information in these methods. Could the proposed method further enhance the performance of ICE-CAM?

[R1] Domain-specific batch normalization for unsupervised domain adaptation, CVPR, 2019.

[R2] Batch normalization embeddings for deep domain generalization, Pattern Recognition, 2023.

**Questions:**

1.As similar feature normalization techniques [R1, R2] have been extensively explored in domain generalization (DG) and domain adaptation (DA) tasks, my main concern is that the novelty of this paper may be limited. It would be beneficial to highlight the advantages of this paper compared to existing DG methods for clearer presentation.

2.In my opinion, the effect of the proposed method is similar to that of domain generalization (DG) methods. Therefore, it is not appropriate to directly compare it with supervised or unsupervised methods in the DG setting (training on MSMT and testing on other datasets). For a more thorough comparison, it would be helpful to include additional related DG and DA methods in the experiments.

3.In Table 3, it shows that the improvement achieved by combining the proposed method with the supervised person ReID task (highlighted in red) is limited. More analysis is needed to explain the reasons for this limitation in this context.

4.Since ICE-CAM and PPLR-CAM have demonstrated that unsupervised person ReID methods can benefit from camera information, the improvements observed in the unsupervised ReID setting (highlighted in gray) may be due to the incorporation of camera information in these methods. Could the proposed method further enhance the performance of ICE-CAM?

[R1] Domain-specific batch normalization for unsupervised domain adaptation, CVPR, 2019.

[R2] Batch normalization embeddings for deep domain generalization, Pattern Recognition, 2023.

---

> ### Author Response · Authors · 2024-11-21
> **Response to Reviewer ffre**
>
> We sincerely appreciate your constructive feedback on our work. We have carefully reviewed your concerns and prepared the following responses to address them. We hope these responses provide clarity and prove helpful. If there are any remaining issues or areas requiring further discussion, we would be glad to engage further. Once again, thank you for reviewing our paper and contributing to its improvement.

---

> ### Author Response · Authors · 2024-11-21
> **1. Differences from previous works and our contributions**
>
> As pointed out, normalization techniques have been employed for domain generalization (DG) and domain adaptation (DA) tasks [R1, R2]. However, our work differs from [R1, R2] in the research domain and application method. Specifically, we focus on the Person ReID problem and utilize normalization to address the camera bias of ReID models. Unlike prior works that propose network layers requiring training, our approach is a post-processing method applied to embedding vectors at test time, making it applicable to arbitrary pretrained models.
>
> As our primary contribution, we highlight the comprehensive analyses of the camera bias of ReID models and the insights derived from these studies. In this work, we try to understand the bias of the existing ReID models. We quantitatively investigate the camera bias of the ReID models in the context of data distribution shifts, which has not been thoroughly explored. Additionally, we revisit normalization as a debiasing method for ReID models and analyze why normalization can mitigate the camera bias through empirical studies, showing its generalizability across various models and benchmarks. Furthermore, we explore its applicability to detailed bias factors such as low-level image properties and body angles. Beyond normalization, we provide insightful analyses of the risks associated with camera bias in unsupervised learning for ReID models, as discussed in Section 5. We believe these extensive analyses and insights into camera bias will offer a valuable contribution for advancing future research in Person ReID.
>
> [R1] Chang et al., Domain-specific batch normalization for unsupervised domain adaptation, CVPR 2019.
> [R2] Segu et al., Batch normalization embeddings for deep domain generalization, Pattern Recognition 2023.

---

> ### Author Response · Authors · 2024-11-21
> **2. Results of DG and DA methods**
>
> As we are dealing with the data distribution shift, the result using DG and DA methods would be helpful. In fact, Table 3 of the main paper includes the results of applying the camera-specific normalization to the state-of-the-art DG ReID methods, ISR (Dou et al., 2023) and PAT (Ni et al., 2023). ISR is trained on a large amount of external video data and PAT is trained on MSMT. In addition, we provide the result using an unsupervised DA method, SPCL-UDA (Ge et al., 2020), which is a variant of SPCL (Ge et al., 2020). The model is trained on Market-1501 (source domain) and MSMT17 (target domain). The pretrained weights released by the original authors were used in the experiment. We summarize the mAP evauation result of the aformentioned DG and DA methods as follows:
> ISR:
> | Features             | Market-1501 | MSMT17 | CUHK03-NP | PersonX |
> |---------------------|-------------|--------|-----------|---------|
> | Original |   70.2      |  32.5   |  38.6    |  66.4   |
> | Normalized |   71.9    | 34.2  |   42.3   |  70.2  |
>
> PAT:
> | Features             | Market-1501 | MSMT17 | CUHK03-NP | PersonX |
> |---------------------|-------------|--------|-----------|---------|
> | Original |    43.8     | 54.8   |  24.5    |  50.0   |
> | Normalized |  52.9    | 54.1   |  29.8    |  59.8  |
>
> SPCL-UDA:
> | Features             | Market-1501 | MSMT17 | CUHK03-NP | PersonX |
> |---------------------|-------------|--------|-----------|---------|
> | Original |   86.8      |   26.8  |  13.9    |  36.1   |
> | Normalized |   86.1    | 28.5   |  18.9    |  45.4  |
>
> All of them exhibit performance gains on their unseen domains with the camera-speicfic normalization. PAT and SPCL-UDA show limited improvement on their supervised domains, MSMT17 and Market-1501, respectively. It is likely due to the fact that they have already less bias towards the cameras of their seen domains through the supervised learning. This relates to the next paragraph "3. Limited improvement of supervised models on seen domain" below. The result of SPCL-UDA has been added in Appendix G.

---

> ### Author Response · Authors · 2024-11-21
> **3. Limited improvement of supervised models on seen domain**
>
> In Table 3 of the main paper, the supervised models exhibit limited performance improvement through the normalization when given seen domain data. This is likely because the models are already less biased towards the cameras of the seen domain, owing to the supervised learning conducted with data from that domain. For example, SOLIDER shows the camera bias value of 21.3, which is almost as small as the GT (the camera imbalance of the dataset) value of 19.2. In addition, since TransReID is learned in a camera-aware manner and leverages the camera labels as side information at inference time, the role of the camera-specific normalization for reducing the camera bias may be redundant. However, it is worth noting that the performance of the supervised models on unseen domains is significantly enhanced, implying that the normalization method clearly improves the generalization ability of the models.

---

> ### Author Response · Authors · 2024-11-21
> **4. Result of ICE-CAM**
>
> We provide the evaluation result of the camera-specific normalization for ICE-CAM (Chen et al., 2021) trained on MSMT. The model was reproduced by us since there were no public pretrained weights. The measured mAP performance is as follows:
> | Features             | Market-1501 | MSMT17 | CUHK03-NP | PersonX |
> |---------------------|-------------|--------|-----------|---------|
> | Original |    25.9     |   37.8  |   12.2   |  26.2   |
> | Normalized |   34.9    |   37.9  |   17.2    |  39.8  |
>
> Like other camera-aware unsupervised methods in Table 3 of the main paper, the performance improvement on the seen domain (MSMT17) is relatively limited. This is likely because the model had already learned the camera information of that domain during the camera-aware training. As a result, the camera-related information provided by the normalization might not offer much additional novelty or benefit to the model. However, it is worth noting that the performance on unseen domains improved significantly. This indicates that the camera knowledge learned by the model is not generalized to other domains, which is effectively complemented by the camera-specific normalization. This result has been updated in Table 3 of the main paper.

---

> ### Comment · Reviewer_ffre · 2024-11-22
>
> The authors have addressed most of my concerns, and I will raise my score to 6.

---

### Official Review · Reviewer_Grbk · 2024-11-02

**Soundness:** 4
**Presentation:** 4
**Contribution:** 4
**Rating:** 8
**Confidence:** 5

**Summary:**

This paper discusses camera bias, where it might come from, and a simple normalization solution. In particular, the paper uses a test-time batch to compute the feature mean and variance of each camera and then normalize the features in each camera. The method is shown to be useful across backbones and training methods.

**Strengths:**

- Overall I like this paper. The authors obviously provided lots of interesting insights. They found that existing methods have camera bias, especially for self-supervised/unsupervised ones. The ones that have dedicated camera-aware design has less camera bias. I believe this observation is new.

- The method is simple and effective. While this method has been used in test-time adaptation, authors use it to address a different problem in person re-id, which should be encouraged.

- Rich insights are provided why camera bias arise.

- I particularly like the fact that this method can be used to deal with human viewpoint bias (Sun and Zheng, 2019). Authors clearly have strong understanding of the field.

- The paper is well written.

**Weaknesses:**

I don't have particular problem with this paper. If I'm to write this paper, perhaps I will improve the structure a bit more. But given that this paper talks about observations, insights, and an easy solution, perhaps the current struture is fine.

Reviewers may have a problem with novelty. But I would champion this way of doing research - there is no need to come up with some new method if the paper is sound and insightful in itself.

**Questions:**

None.

---

> ### Author Response · Authors · 2024-11-21
> **Response to Reviewer Grbk**
>
> We sincerely appreciate the positive comments and recognition of the contributions of our work. We believe that this paper provides valuable insights for understanding camera bias of person ReID models. We will think about a better paper structure. The additional experiments and analyses conducted based on the comments of the other reviewers have been incorporated into the paper, so we would appreciate it if you could keep an eye out for the updates. Once again, thank you for your commitment to reviewing and for your encouragement.

---

> > ### Comment · Reviewer_Grbk · 2024-11-21
> > **I think the re-id field needs such papers**
> >
> > Good work. I look forward to seeing more such papers.

---

### Official Review · Reviewer_fCPa · 2024-11-04

**Soundness:** 3
**Presentation:** 3
**Contribution:** 2
**Rating:** 8
**Confidence:** 5

**Summary:**

This paper empirically investigates camera bias in person re-identification (ReID) models. The authors measure camera bias in ReID models across unseen domains, demonstrating its amplification under data distribution shifts. They analyze the role of feature normalization in mitigating bias, showing it effectively addresses specific bias factors like low-level image attributes and body angles. Furthermore, observing the negative effect of camera-biased pseudo labels on training, the paper proposes simple training strategies to alleviate this bias.

**Strengths:**

The paper is well-written, with clear motivation and visual aids enhancing the analysis. The main contributions of this work are as follows:
1. The authors investigate camera bias in ReID models across unseen domain data, offering a comprehensive analysis that spans diverse learning methods and model architectures.
2. The authors' analysis explains why it is effective for bias mitigation and shows its applicability to detailed bias factors and multiple models.
3. The authors explore the risk of camera bias inherent in unsupervised learning of ReID models, showing that the performance of existing unsupervised algorithms can be effectively enhanced by simple modifications to reduce the risk.

**Weaknesses:**

1. The authors’ analysis lacks consideration of the spatial distribution of features, such as a scatterplot of features downscaled to two dimensions using t-SNE. This visualization could reveal feature distributions under camera bias conditions.
2. The paper primarily addresses domain differences as camera biases. However, another type of camera bias, i.e., the number of samples varies between different cameras, and the identity labels may also be different, is not mentioned in the paper. The authors are welcome to discuss this bias.
3. The description of the training strategy in 5.3 is too simple. Does discarding bias clustering drastically reduce the training samples? In addition, the parameter settings of the clustering algorithm can affect the clustering results, and whether there is a correlation between the settings of these parameters and the camera bias. This is not discussed by the authors

**Questions:**

See Weaknesses.

---

> ### Author Response · Authors · 2024-11-21
> **Response to Reviewer fCPa**
>
> We sincerely appreciate for your review of our work and the valuable suggestions for further analysis. We hope you will find  the below responses helpful. All experiments and discussions from your comments have been incorporated into the paper, which have contributed to the improvement of our work. If you have any further comments or advice, we would be happy to discuss them. Once again, thank you for reviewing our work and contributing to its improvement.

---

> ### Author Response · Authors · 2024-11-21
> **1. Feature visualization**
>
> We provide the t-SNE result of features of a camera-aware model, PPLR-CAM (Cho et al., 2022), trained on the MSMT dataset using samples from the Market dataset in Figure 15. From the left plot, it is clearly observed that the features from the same camera tend to cluster more than the features from the different cameras. This camera bias is effectively mitigated by the normalization as shown in the right plot. This discussion has been updated in Appendix F.

---

> ### Author Response · Authors · 2024-11-21
> **2. Camera imbalance of training data**
>
> The number of samples in typical ReID datasets varies between different cameras and different identities. This is a noteworthy point, as this imbalance might be related to the camera bias of the models. Consequently, if the inherent imbalance of a dataset is corrected, would the camera bias be resolved? In fact, the result of a related experiment had been presented in Appendix E, which has been refined. Here, we summarize it for your convenience.
>
> To find out the aformentioned question, a ReID model (Dai et al., 2022) is trained in a supervised manner on PersonX, where all identities have the same number of samples for each camera. The mAP evaluation result of the camera-specific normalization on the trained model is as follows:
> | Features             | Market-1501 | MSMT17 | CUHK03-NP | PersonX |
> |---------------------|-------------|--------|-----------|---------|
> | Original | 12.7        | 1.2    | 4.5       | 87.8    |
> | Normalized | 18.8        | 2.3    | 7.0       | 88.8    |
>
> It is observed that the effect of the normalization is still definite, indicating that the camera bias persists even in the absence of camera imbalance and identity imbalance in the training data. In other words, more effort is required for debiasing beyond balancing the training dataset.

---

> ### Author Response · Authors · 2024-11-21
> **3. Number of discarded training samples**
>
> We provide the examination result about the ratio of discarded samples (i.e., the samples of single-camera clusters) during the training of CC (Dai et al., 2022) under our training strategy. The result is as follows:
>
> | Epoch             | 0     | 20   | 40   | 60   | 80   | 100  |
> |-------------------|-------|------|------|------|------|------|
> | Discarded samples | 63.5% | 5.8% | 3.0% | 2.6% | 2.8% | 2.8% |
>
> As shown in the table, a large number of training samples is discarded at the initial epoch. However, this proportion subsequently diminishes; only approximately 3% of the total samples are excluded finally. As a result, the risk of camera bias in early training stages is effectively reduced by the large exclusion of samples, and the model learns sufficient knowledge from almost all samples as it converges. We emphasize that, despite its simplicity, the proposed learning strategy is an effective and easy-to-implement solution which considerably enhances the mAP of this model by 19.3%p. This analysis has been added in Appendix H.

---

> ### Author Response · Authors · 2024-11-21
> **4. Influence of clustering parameter**
>
> Since the clustering result depends on the parameter settings of the clustering algorithm, it is possible that the camera bias of the model also varies. We provide an analysis result to examine this issue. Specifically, we investigate the influence of the most important parameter of DBSCAN, Eps, which is the maximum distance between two samples to be neighborhood. The mAP performance and camera bias of the CC models trained on MSMT with Eps values of 0.4, 0.5, 0.6, 0.7, and 0.8 are as follows:
> | Eps    | 0.4  | 0.5  | 0.6  | 0.7  | 0.8 |
> |-----------|------|------|------|------|-----|
> | mAP        | 18.3 | 21.8 | 29.8 | 32.0 | 8.2 |
> | Camera bias | 33.7 | 32.0 | 32.5 | 30.5 | 40.7    |
>
> The larger Eps values tend to roughly decrease the camera bias, while the too large value (Eps = 0.8) results in the severe performance degradation. The number of clusters generated at the last training epoch for each model is as follows:
> | Eps    | 0.4  | 0.5  | 0.6  | 0.7  | 0.8 |
> |-----------|------|------|------|------|-----|
> | # training clusters | 2495 | 2090 | 1564 | 1108 | 291 |
>
> A larger Eps value can make it easier for samples to cluster together, leading to fewer clusters and larger cluster sizes. Thus, it seems that as the Eps value increases, the diversity of cameras in each cluster benefits from the increased cluster size, up to a certain point. However, the samples are indiscriminately clustered for a too large value, causing poor clustering quality. In this context, setting an appropriate value for Eps is crucial for model training. Here, we find that Eps = 0.7 yields the best result, with the number of clusters (1108) most similar to the number of identities (1041) in the training data. In addition, we observe the significant performance improvement of the models from the result of our training strategies:
> | Eps    | 0.4  | 0.5  | 0.6  | 0.7  | 0.8 |
> |-----------|------|------|------|------|-----|
> | mAP        | 44.5 | 46.9 | 49.1 | 46.2 |  -  |
> | Camera bias | 24.6 | 24.4 | 23.8 | 24.0 |   -  |
>
> The result of Eps = 0.8 is omitted since the number of the generated clusters was extremely low, so the mini-batches were not produced properly with the same data sampler used in the previous experiments. This analysis has been updated in Appendix I.

---

> ### Comment · Reviewer_fCPa · 2024-11-22
>
> Thanks for the author's response, I'll raise my score to 8.

---

### Official Review · Reviewer_UN8K · 2024-11-05

**Soundness:** 3
**Presentation:** 4
**Contribution:** 3
**Rating:** 8
**Confidence:** 4

**Summary:**

This paper proposes to measure the camera bias and develop a simple normalizaiton methods to address the affects of such bias. Experiments also shows the effectiveness of the proposed method.

**Strengths:**

1. A thorough investigation of camera bias in reID is a big contribution. This paper provides a quantitative view for bias measuring, which is most welcomed.
2. The consequently proposed debiasing normalization is very simple yet effective, with sufficient experimental and theorical analysis.
3. The new findings could advanced exisiting reID methods.

**Weaknesses:**

1. The adopted reID method is not new, missing some newly proposed methods.
2. If the number of cameras arise to a big one, e.g., 10k in a region, would the bias disappeared? Or the bias could be supressed by multi-view learning?

**Questions:**

See weaknesses.

---

> ### Author Response · Authors · 2024-11-21
> **Response to Reviewer UN8K**
>
> We sincerely appreciate your recognition of our contribution to the ReID field. We hope you find the following answers to the raised questions useful. We would be happy to discuss further if you have any other questions or concerns. Once again, thank you for reviewing our paper and for your valuable feedback.

---

> > ### Comment · Reviewer_UN8K · 2024-11-25
> >
> > Thanks for the author's response. Most of my questions are addressed. I will keep the initial rating of 8.

---

> ### Author Response · Authors · 2024-11-21
> **1. Result of a recent ReID method**
>
> We provide the evaluation result of the camera-specific normalization for a recent camera-aware unsupervised ReID method, CAJ (Chen et al., 2024). The pretrained weights trained on MSMT17 by the original authors were used in the experiment. The result has been updated in Table 3 of the main paper. The mAP performance is summarized as follows:
> | Features             | Market-1501 | MSMT17 | CUHK03-NP | PersonX |
> |---------------------|-------------|--------|-----------|---------|
> | Original |   30.6      |  44.3   |  14.1    |  32.5   |
> | Normalized |  36.9     |  42.8  |  18.1    | 40.9   |
>
> Similar to other camera-aware unsupervised methods, the performance improvement on the seen domain (MSMT17) is limited. This can be attributed to the fact that the model had already acquired the camera-related information of that domain in its camera-aware learning process. Therefore, the camera information from the normalization might not be novel and beneficial for the model. Nevertheless, we note that the performance on the unseen domains is markedly enhanced, suggesting that the normalization method effectively supplements the insufficiently generalized camera knowledge of the model.
>
> [Reference]
> Chen et al., CA-Jaccard: Camera-aware jaccard distance for person re-identification, CVPR 2024.

---

> ### Author Response · Authors · 2024-11-21
> **2. Number of camera and camera bias**
>
> Thanks for raising a question worth thinking about. Given that samples for each identity are evenly distributed across all cameras, a very large number of cameras in a training dataset would be significantly helpful for reducing camera bias. However, this may not be enough to make the camera bias totally disappear. In our opinion, the ambiguity inherent in ReID training (e.g., hard positives and hard negatives due to changes in visible body parts, occlusion, etc.) can fundamentally make a ReID model susceptible to capturing noise other than identity information from the images during training. However, to be free from the camera bias, a model must be able to extract the identity information from arbitrary images while completely ignoring camera information such as low-level properties, background, body angle, etc. Training a model to do this may require more refined datasets and sophisticated learning strategies, and the multi-view learning also might help.

---

### Author Response · Authors · 2024-11-21
**Revision**

We sincerely appreciate the valuable feedback from all reviewers. We have thoroughly reviewed the comments and provided detailed responses to clarify our statements and offer additional analysis results. The updates made to the paper are summarized as follows:
- The evaluation results of ICE-CAM (Chen et al., 2021) and CAJ (Chen et al., 2024) have been added in Table 3.
- The investigation on the influence of camera balance of training data has been revised in Appendix E.
- The feature visualization result has been updated in Appendix F.
- The evaluation result of an unsupervised domain adaptation method has been updated in Appendix G.
- The analysis on the number of discarded training samples by our training strategy for USL has been updated in Appendix H.
- The experiment result of varying a clustering parameter in USL has been updated in Appendix I.
- Minor typos have been corrected.

Once again, we thank the reviewers for their constructive feedback, which has greatly helped us in further refining our work.

---

### Meta-Review · Area_Chair_3Bz2 · 2024-12-20

**Metareview:**

This paper received four positive ratings, with all reviewers generally inclined to accept it. According to the reviews, the paper is well-written, with a clear motivation and visual aids that effectively enhance the analysis. The authors explore camera bias in ReID models across unseen domain data, providing a comprehensive analysis that covers various learning methods and model architectures. All reviewers appreciate the in-depth examination of camera bias, and they find the proposed method both simple and effective, supported by adequate experimental and theoretical analysis. Furthermore, all weaknesses have been addressed in the authors' response. Overall, it is a good work, and the Area Chair (AC) recommends accepting the paper.

**Additional Comments On Reviewer Discussion:**

This paper received four positive ratings, with all reviewers generally inclined to accept it. Furthermore, all weaknesses have been addressed in the authors' response.

---

### Decision · Program_Chairs · 2025-01-22

Accept (Spotlight)